# Assessing the Knowledge-intensive Reasoning Capability of Large Language Models with Realistic Benchmarks Generated Programmatically at Scale

## Abstract

Although LLMs demonstrate strong reasoning capability in such tasks as mathematical problem solving, less is known about their reasoning capability in settings that require extensive real-world knowledge due to the limited scale and knowledge coverage of existing benchmarks. To shed more light into this, we propose a novel pipeline that is capable of programmatically generating realistic knowledge-intensive question answering benchmarks that require complex reasoning. Leveraging open knowledge graphs, the graph query language SPARQL, and LLMs, our pipeline requires no manual annotation and can therefore scale to unprecedented benchmark size and knowledge coverage. We evaluate several state-of-the-art LLMs with benchmarks generated by our pipeline, and find that the LLMs struggle to recall and leverage world knowledge for reasoning, even for world knowledge present in their pre-training corpuses. Additionally, retrieval-augmented generation and chain-of-thoughts prompting does not fully solve the problems. Our benchmarks further enable us to examine to what extent the confidence of LLMs in the outcomes of their reasoning transparently reflects their confidence in the underlying knowledge, a study that is first-of-its-kind to our best knowledge. We find that the confidence of LLMs in the outcomes of their reasoning reflects poorly their confidence in the underlying knowledge (poor knowledgeability transparency), which suggests a direction of future improvement.

## 1 Introduction

Existing benchmarks examining how well LLMs can leverage real-world knowledge for reasoning predominantly rely on human annotations, e.g. Yang et al. (2018); Kwiatkowski et al. (2019). The costs of manual annotations limit the scale of these benchmarks even in unspecialized domains where manual annotations are affordable, let alone domains where shortages of domain experts make manual annotations prohibitively costly (Hendrycks et al., 2020). The reliance of existing approaches on human annotations also limits the coverage of long-tail knowledge by these benchmarks, since such knowledge is often beyond the expertise of most human annotators. We propose a pipeline for generating such benchmarks that is fully automated and therefore much more scalable. Given a KG, our pipeline first samples subgraphs from the KG, masks a subset of entities in the subgraphs, and encodes the masked subgraphs as SPARQL queries. Our pipeline then translate the SPARQL queries into natural language questions by an LLM, and obtain ground truth answers to the LLM-generated questions by querying the KG.

With benchmarks generated by our pipeline, we further assess to what extent SOTA LLMs can recall and leverage world knowledge for complex reasoning, in the setting of zero-shot question answering, retrieval-agumented generation (RAG), and chain-of-thoughts (CoT) prompting. Our assessment yields the following findings:

1. LLMs have significant room of improvement when it comes to recall and leverage world knowledge for reasoing, even for world knowledge present in their pre-training corpuses.

2. LLMs cannot avoid reasoning errors even with all required knowledge provided as context.

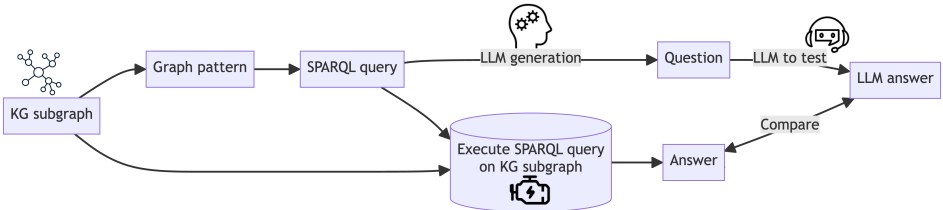

Figure 1: Our assessment scheme

*Subgraph*:
<Surround SCM> <developer> <Seapine Software> .
<Surround SCM> <operating system> <Microsoft Windows> .
<Surround SCM> <has use> <version control>

*Graph pattern*:
?x <developer> <Seapine Software> .
?x <operating system> <Microsoft Windows> .
?x <has use> <version control>

(a) Step 1: Sample a subgraph (above) in the knowledge graph and turn it into a graph pattern (below) by masking some of the entities in it.

**User**: Please translate the following SPARQL query into a wh-question:
SELECT ?x WHERE {
?x <developer> <Seapine Software> .
?x <operating system> <Microsoft Windows> .
?x <has use> <version control>
}
**Assistant**: What is developed by Seapine Software, uses Microsoft Windows as operating system, and utilizes version control?

(b) Step 2: Translate the SPARQL query consisting of the graph pattern into a natural language question with an LLM.

Figure 2: Our benchmark generation pipeline by example

3. CoT improves performance for complex questions, but such improvement can be unstable, which can be further worsened by mismatches between the reasoning pattern of demonstration and test question.

4. LLMs' confidence (as measured by log-likelihood) in their answers reflects poorly their confidence in the underlying knowledge (poor knowledgeability transparency).

The rest of the paper is organized as the following: we introduce the methodology of our assessment in Section 2, introduce the concept of knowledgeability transparency in Section 3, present the findings of our assessment in Section 4, and review related work in Section 5.

## 2 ASSESSMENT METHODOLOGY

### 2.1 BACKGROUND

Question answering is a natural way to assess knowledge-intensive reasoning. Existing approaches to building complex question answering benchmarks predominantly require significant manual annotations. The most labor-intensive approach requires human annotators to raise questions about pieces of text. Although they yield highly diverse questions, such approaches can have prohibitive costs. The lack of structure in even relatively organized texts such as Wikipedia makes it hard to quantify the reasoning capabilities that are measured. For example, a study Min et al. (2019) finds that a surprisingly large portion of questions in HotpotQA Yang et al. (2018), a popular multi-hop question answering dataset, can be answered without multi-hop reasoning. Less labor intensive approaches often rely on customized domain-specific language as intermediate representation (IR). Constructing such IRs itself can also require significant manual efforts. Additionally, customized IRs may not generalize well to novel knowledge. Finally, these approaches generally rely on manual efforts to paraphrase questions expressed IRs into natural language. Those approaches in this category that does not rely on manual paraphrasing instead rely on manually crafted templates, which can be costly to construct. This paper demonstrates that graph query languages such as SPARQL can act as powerful and general IRs readily digestible for LLMs-empowered question answering benchmark

| KG | Entities | Predicates | Triples | Alignment |
|---|---|---|---|---|
| T-REx | 2,819,966 | 658 | 5,410,928 | Wikipedia |
| Wikidata | 11,718,022 | 1428 | 50,732,257 | N/A |

Table 1: Statistics of KGs used for benchmark generation

| Dataset | Generation | Source(s) | Size |
|---|---|---|---|
| WikiHop (Welbl et al., 2018) | Template | Wikidata | 51,318 |
| Natural Questions (Kwiatkowski et al., 2019) | Manual | Wikipedia | 323,044 |
| HotpotQA (Yang et al., 2018) | Manual | Wikipedia | 113k |
| ComplexWebQuestions (Talmor and Berant, 2018) | Paraphrasing | Freebase | 34,689 |
| CFQ (Keysers et al., 2020) | Template | Freebase | 865,101 |
| Quest (Malaviya et al., 2023) | Paraphrasing | Wikipedia | 3,357 |
| **Ours** | LLM | Wikidata/T-REx | 1.32M |

Table 2: Comparison of datasets. Note that datasets generated from templates may not have realistic natural language form, and datasets generated manually or by paraphrasing can be hard to scale.

generation that requires no manual efforts. We provide more details of datasets generated by existing approaches in Table 2.

## 2.2 LLM-EMPOWERED BENCHMARK GENERATION WITH SPARQL AS IR

Our benchmark generation pipeline takes as input a knowledge graph (KG), which we define as a collection of semantic triples $\mathcal{KG} = \{(s, r, o)\} \subset \mathcal{E} \times \mathcal{P} \times (\mathcal{E} \cup \mathcal{V})$, where $\mathcal{E}$ is a set of entities, $\mathcal{P}$ is a set of predicates, and $\mathcal{V}$ is a set of values. To generate a question that requires reasoning, we first sample a subgraph in the KG by taking a random walk (see Algorithm 1 for the subgraph sampling procedure and Figure 2a for an example subgraph in Wikidata). and a subset of the entities in the subgraph are masked (see Figure 2a for an example). We mask some of the entities in the sampled subgraph, and encode the partially masked subgraph as a graph query in SPARQL, a widely adopted graph query language endorsed by W3C. We then translate the SPARQL query into a natural language question with an LLM. We choose SPARQL because we find state-of-the-art LLMs, such as GPT-4, demonstrates strong capabilities of translating SPARQL queries into natural language questions. Finally, we find the set of ground truth answers to the question by executing the SPARQL query against the KG. We show an example of this process in Figure 2.

---
**Algorithm 1** Subgraph sampling procedure

---
**Require:** Knowledge graph $\mathcal{KG} = \{(s, r, o)\}$, subgraph size $n$, returning probability $p$
   $\mathcal{G} \leftarrow \{\}$              ▷ Empty subgraph
   $s, \_, \_ \sim (\mathcal{KG})$              ▷ Start of random walk
   **while** $|\mathcal{G}| < n$ **do**
      $s, r, o \sim (\{(s, r, o) : (s, r, o) \in \mathcal{KG}\})$     ▷ Sample a semantic triple
      $\mathcal{G} \leftarrow \mathcal{G} \cup \{(s, r, o)\}$      ▷ Add the sampled triple to the subgraph
      **if** rand() $< p$ **then**     ▷ With probability $1 - p$ stay at $s$ in the next iteration
         $s \leftarrow s'$
   **if** nontrivial($\mathcal{G}$) **then return** $\mathcal{G}$ ▷ Return the masked subgraph only if the query is nontrivial

---

## 2.3 CHOICE OF KNOWLEDGE GRAPHS

Apart from scale and quality, our primary concern when choosing KGs is their alignment with the pre-training corpuses of SOTA LLMs. Based on this, we choose to generate benchmarks from the T-REx and Wikidata. We report key statistics of Wikidata and T-REx in Table 1. Both KGs can contain offensive contents. Our pipeline cannot filter such contents.

**T-REx** ElSahar et al. (2018) (CC BY-SA 4.0 DEED) is a large scale knowledge graph generated by aligning Wikipedia paragraphs. Since it is aligned to Wikipedia, it has strong guarantee that all facts contained by it are contained in the pre-training corpus of LLMs.

**Wikidata** Vrandečić and Krötzsch (2014) (CC BY-SA 4.0 DEED) is a community maintained KG managing facts in Wikipedia, its sister project. To maximize its overlap with the pre-training corpuses of LLMs that we evaluate, we only include Wikidata entities with English Wikipedia pages associated to them. Despite this, there is no guarantee that facts in Wikidata can be backed by any Wikipedia page. Note that T-REx is not a proper subset of Wikidata.

## 2.4 METHODOLOGY LIMITATIONS

**Exploitable correlations** Knowledge graphs usually contain facts that are highly correlated to each other, which LLMs can exploit to bypass recalling facts. For example, the predicates "country for sport" and "country of citizenship" in Wikidata are highly likely to co-occur. Consequently, queries with graph patterns such as "`?x0 <country for sport> <United States> . ?x0 <country of citizenship> ?x1`" can be answered by LLMs with high accuracy without recalling any athletes that are member of a US sport team and hold US citizenship. We leave it for future work to filter such correlated predicates in our benchmark generation pipeline. In this case, however, the question still requires some commonsense reasoning.

**Ground truth incompleteness** LLMs can generate answers that are factually correct but may not be included in the ground truth sets of our benchmarks. We find that LLMs' performance is comparable on the benchmark generated from T-REx with that generated from Wikidata, even though Wikidata contains about 10 times more triples than T-REx and therefore is much less likely to suffer from ground truth incompleteness. This suggests that ground truth incompleteness may not be a foundational problem for our methodology. However, it will be worth studying further the impact of this.

## 3 KNOWLEDGEABILITY TRANSPARENCY

**Definition** We consider an LLM $M$ to have knowledgeability transparency if its confidence in its answers is proportional to its knowledgeability about questions. We define this mathematically as, given a question $q$ and its answer $a$,

$$\log \Pr[a|q, M] \propto K_M(q, a) \tag{1}$$

where $K_M(q, a)$ denotes the knowledgeability of model $M$. Since we are primarily interested in employing LLMs to answer questions generatively, all distributions $\Pr[\cdot|\cdot]$ are text generation distribution. For our benchmarks, where each question covers multiple facts, at least two definitions of knowledgeability can be considered:

$$K_{\text{sum}}(q, a) = \sum_{f \in F(q,a)} \log \Pr[a|f_{\backslash a}, M] \qquad K_{\text{min}}(q, a) = \min_{f \in F(q,a)} \log \Pr[a|f_{\backslash a}, M]$$

where $F(q, a)$ denotes the set of facts that are required to conclude that $a$ is a correct answer of $q$, and $f_{\backslash a}$ denotes the question for what entities the fact $f$ holds. For example, for the fact `<George Washington> <president> <United States>` and the answer `<George Washington>`, the question $f_{\backslash a}$ could be "Who was a president of the United States?"

**Limitation and alternative** The number of correct answers can lead to unexpected fluctuations of quantities in Definition 1. For example, consider the question "Who participated in both event A and B?", and suppose that the event A and B have two participants each and one participant X in common. Since both events have two participants, an LLM that knows both events well is likely yield

$$\log \Pr[\text{"X"}|\text{"Who participated in event A?"}, M] \approx \log \Pr[\text{"X"}|\text{"}\dots\text{in event B?"}, M] \leq \log 1/2$$

Since the two events only have one common participant X, the same LLM is likely to yield

$$\log \Pr[\text{"X"}|\text{"Who participated in both event A and B?"}, M] \approx \log 1$$

Consequently, $\log \Pr[a|q, M] = 1 > K_M(q, a)$, making the LLM appear overconfident in its knowledgeability while it actually is not.

As an alternative, we can replace $\log \Pr[a|q, M]$ with $\log \Pr[Y|\bar{q}_a, M]$, where $\bar{q}_a$ denotes the question if $a$ is a correct answer to $q$, and $T$ is some token expressing affirmation, such as "Yes", e.g. $\log \Pr["Yes"|"Did X participate in both event A and B?", M]$. We can similarly replace $\log \Pr[a|f_{\backslash a}, M]$ with $\log \Pr[Y|\bar{f}_a, M]$, e.g. $\log \Pr["Yes"|"Did X participate in event A?", M]$. This results in the following metrics of knowledgeability:

$$\bar{K}_{\text{sum}}(q, a) = \sum_{f \in F(q,a)} \log \Pr[Y|\bar{f}_a, M] \qquad \bar{K}_{\text{min}}(q, a) = \min_{f \in F(q,a)} \log \Pr[Y|\bar{f}_a, M] \qquad (2)$$

Importantly, the question $\bar{q}_a$ and $\bar{f}_a$ can both be generated by LLMs from underlying SPARQL queries, similar to the way the question $q$ is generated. This makes these metrics feasible to evaluate.

## 4 EXPERIMENTS

### 4.1 SETTINGS

With T-REx and Wikidata, we generate questions from SPARQL queries with 1, 2, and 3 unknowns in their graph patterns. Queries with 1 unknown have graphs patterns of size from 2 to 6. Queries with 2 and 3 unknowns have graph patterns of size 4, 5, and 6. For each number of unknowns and graph pattern size, we generate 100k natural language questions with Llama-3-70B-Instruct.

**Data quality** To ensure the accuracy of the LLM translations of SPARQL queries, we manually examined 1,200 questions (400 questions for each number of unknowns). We report the sample and 95% lower confidence bound of the translation accuracies in Table 3.

**Answer verification** To account for explanatory texts that LLMs often generate when answering questions, we deem an LLM answer correct if any ground truth answer matches exactly some part of the answer. We lower-case and remove accents from LLM and ground truth answers before matching.

| $n_{\text{unk}} = 1$ | | $n_{\text{unk}} = 2$ | | $n_{\text{unk}} = 3$ | |
|---|---|---|---|---|---|
| Acc | LCB | Acc | LCB | Acc | LCB |
| 94% | 89.4% | 91% | 85.4% | 87% | 80.4% |

Table 3: Sample and 95% lower confidence bound (LCB) of SPARQL translation accuracy.

### 4.2 ZERO-SHOT QUESTION ANSWERING

We first evaluate the zero-shot question answering accuracy of several SOTA LLMs and study the impact of knowledgeability on their zero-shot accuracy. For questions with one unknown, we report the zero-shot question answering accuracy in Table 4 (Wikidata) and Table 5 (T-REx), grouped by graph pattern size. For questions with 2 and 3 unknown, we report the zero-shot question answering accuracy in Table 6 (Wikidata) and Table 7 (T-REx), grouped by the number of unknowns and graph pattern size. We use 100k questions to compute the accuracy of all models except those in the GPT family, whose accuracy are computed with 5k questions, sampled randomly from the 100k questions. We also show in Figure 3 a hallucinative answer from GPT-4. Because all facts in T-REx are aligned

| Model | $|\mathcal{G}| = 2$ | $|\mathcal{G}| = 3$ | $|\mathcal{G}| = 4$ | $|\mathcal{G}| = 5$ | $|\mathcal{G}| = 6$ |
|---|---|---|---|---|---|
| Llama-3-70B-Instruct | 28.8 | 21.2 | 16.6 | 14.3 | 14.0 |
| Llama-3-8B-Instruct | 19.6 | 13.6 | 9.6 | 7.2 | 6.3 |
| Mixtral-8x7B-Instruct-v0.1 | 22.5 | 16.5 | 12.2 | 9.9 | 9.4 |
| gpt-3.5-turbo-0125 | 24.7 | 18.3 | 15.0 | 11.1 | 10.4 |
| gpt-4o-2024-05-13 | 32.2 | 23.8 | 20.0 | 18.1 | 17.8 |

Table 4: Zero-shot question answering accuracy for questions generated from Wikidata with different graph pattern sizes and one unknown.

| Model | $|\mathcal{G}| = 2$ | $|\mathcal{G}| = 3$ | $|\mathcal{G}| = 4$ | $|\mathcal{G}| = 5$ | $|\mathcal{G}| = 6$ |
|---|---|---|---|---|---|
| Llama-3-70B-Instruct | 34.8 | 24.4 | 23.1 | 28.4 | 36.1 |
| Llama-3-8B-Instruct | 23.0 | 14.0 | 11.7 | 14.0 | 18.4 |
| Mixtral-8x7B-Instruct-v0.1 | 27.7 | 18.1 | 16.2 | 19.4 | 25.0 |
| gpt-3.5-turbo-0125 | 30.3 | 20.5 | 20.8 | 24.8 | 32.5 |
| gpt-4o-2024-05-13 | 38.7 | 28.4 | 28.9 | 34.8 | 45.3 |

Table 5: Zero-shot question answering accuracy for questions generated from T-REx with different graph pattern sizes and one unknown.

> **User:** What athlete named Karl participated in sailing in the 1980 Summer Olympics?
> **Assistant:** The athlete named Karl who participated in sailing in the 1980 Summer Olympics is Karl Schäfer.

Figure 3: A hallucinative answer from GPT-4 (in red). The only athelete named Karl Schäfer that we find was an Austrian figure skater and swimmer who died in 1976.

with Wikipedia articles, they are also guaranteed to occur in the pre-training corpuses of the LLMs that we evaluate, which suggests:

**Finding 1**: LLMs have significant room of improvement when it comes to recall and leverage world knowledge for reasoing, even for world knowledge present in their pre-training corpuses.

## 4.3 QUESTION ANSWERING WITH RAG

We further evaluate the question answering performance of LLMs when given relevant knowledge. We only use the benchmark generated from T-REx for this experiment, because all facts in T-REx are aligned to Wikipedia articles. (Facts in Wikidata are not guaranteed to be aligned to any sources available publicly, even though they frequently are.) This eliminates the need of stand-alone retrievers, which can introduce cascading errors.

We consider two retrieval settings. In the first setting, we only supply LLMs with Wikipedia articles about entities explicitly mentioned in questions. This setting corresponds to a simple retriever that leverages entity linking and basic information retrieval. In the second setting, in addition to Wikipedia articles supplied in the first setting, we also supply LLMs with Wikipedia articles containing all facts necessary for answering the questions. This setting corresponds to an oracle retriever that always supplies necessary knowledge. In both settings, we use 5k questions with 1 unknown. For gpt-3.5-turbo, we use 1k questions with 1 unknown. We report the results in Tabel 8 (basic retriever) and Tabel 9 (oracle retriever). Despite improvements, a gap persists between the LLMs' performance and its upper bound, which suggest that:

**Finding 2:** LLMs can still make reasoning mistakes even with all knowledge necessary for reasoning.

| Model | $|\mathcal{G}| = 4$ | | $|\mathcal{G}| = 5$ | | $|\mathcal{G}| = 6$ | |
|---|---|---|---|---|---|---|
| | $n_{\text{unk}} = 2$ | $n_{\text{unk}} = 3$ | $n_{\text{unk}} = 2$ | $n_{\text{unk}} = 3$ | $n_{\text{unk}} = 2$ | $n_{\text{unk}} = 3$ |
| Llama-3-70B | 44.1 / 47.7 | 27.7 / 32.1 | 47.0 / 54.3 | 27.3 / 29.8 | 47.7 / 53.0 | 27.2 / 31.9 |
| Llama-3-8B | 40.2 / 34.0 | 23.1 / 18.6 | 43.4 / 37.0 | 23.5 / 9.5 | 44.3 / 45.0 | 23.2 / 11.4 |
| Mixtral-8x7B-v0.1 | 44.7 / 42.9 | 25.3 / 24.9 | 48.1 / 46.7 | 25.2 / 21.2 | 50.3 / 46.0 | 25.0 / 21.7 |
| gpt-3.5-turbo-0125 | 36.0 / 47.9 | 23.4 / 29.5 | 40.7 / 51.8 | 23.4 / 28.3 | 42.4 / 53.2 | 22.2 / 29.1 |

Table 6: Zero-shot and 8-shot CoT accuracy for questions with 2 and 3 unknowns generated from Wikidata. All open-source models are instruction-finetuned version.

| Model | $\|\mathcal{G}\| = 4$ | | $\|\mathcal{G}\| = 5$ | | $\|\mathcal{G}\| = 6$ | |
|---|---|---|---|---|---|---|
| | $n_{unk} = 2$ | $n_{unk} = 3$ | $n_{unk} = 2$ | $n_{unk} = 3$ | $n_{unk} = 2$ | $n_{unk} = 3$ |
| Llama-3-70B | 47.3 / 59.3 | 34.4 / 35.1 | 46.7 / 66.4 | 34.0 / 36.5 | 43.9 / 63.9 | 35.0 / 32.6 |
| Llama-3-8B | 46.8 / 55.4 | 27.7 / 11.8 | 48.0 / 55.1 | 27.1 / 16.2 | 45.8 / 50.9 | 27.6 / 11.5 |
| Mixtral-8x7B-v0.1 | 46.9 / 58.9 | 30.3 / 27.8 | 47.9 / 64.5 | 30.1 / 28.9 | 45.8 / 66.5 | 30.8 / 26.7 |
| gpt-3.5-turbo-0125 | 38.0 / 60.0 | 28.1 / 31.2 | 37.3 / 63.0 | 28.6 / 34.7 | 36.3 / 70.2 | 30.3 / 34.9 |

Table 7: Zero-shot and 8-shot CoT accuracy for questions with 2 and 3 unknowns generated from T-REx. All open-source models are instruction-finetuned version.

| Model | $\|\mathcal{G}\| = 2$ | $\|\mathcal{G}\| = 3$ | $\|\mathcal{G}\| = 4$ | $\|\mathcal{G}\| = 5$ | $\|\mathcal{G}\| = 6$ |
|---|---|---|---|---|---|
| Llama-3-70B-Instruct | 38.8 | 27.5 | 28.4 | 34.9 | 43.1 |
| Llama-3-8B-Instruct | 27.9 | 20.3 | 19.9 | 25.7 | 33.1 |
| Mixtral-8x7B-Instruct-v0.1 | 32.5 | 23.9 | 23.0 | 27.7 | 36.2 |
| gpt-3.5-turbo-0125 | 34.9 | 24.4 | 23.8 | 27.7 | 34.1 |

Table 8: Question answering accuracy with basic retrieval, grouped by graph pattern size $\|\mathcal{G}\|$.

## 4.4 QUESTION ANSWERING WITH CoT PROMPTING

We next study if CoT prompting Wei et al. (2022) can improve the performance of the LLMs. We only use questions with 2 or 3 unknowns for this experiment, since they are harder to answer without chain of reasoning. To generate CoT demonstrations, we employ LLMs to translate semantic triples that encode necessary reasoning steps into natural language statements. An example CoT demonstration thus generated can be found in Figure 4. We use 10k questions except for gpt-3.5-turbo, for which we use 1k questions. We report the question answering accuracy of the LLMs with CoT prompting in Table 6 (Wikidata) and Table 7 (T-REx). We additionally study if the CoT performance can be affected by mismatch between the reasoning pattern of demonstrations and test questions. We report the result in Table 10 (Wikidata) and Table 11 (T-REx). The results suggest:

**Finding 3**: CoT improves performance for complex questions, but it can be unstable, and can be worsened by mismatches between the reasoning pattern of demonstration and test question.

## 4.5 KNOWLEDGEABILITY TRANSPARENCY

We plot in Figure 6a the quantities $\log \Pr[a|q, M]$ and $K_M(q, a)$ (both $K_{\min}$ and $K_{\text{sum}}$) for Llama-3-70B-Instruct as defined in Equation 1. We use 1k question-answer pairs for the plot˙. Although the two quantities are positively correlated, the plot suggests that such metrics indeed struggle from the effect. Further, the correlation may arise from other factors. For example, longer answers may generally have lower log-likelihoods, regardless of questions. We therefore also plot the alternative metrics $\log \Pr[Y|\bar{q}_a, M]$ and $\bar{K}_{\min}$ and $\bar{K}_{\text{sum}}$ in Figure 6b and Figure 6c. These plots indicate very poor correlation between the quantities.

We further study the impact of the knowledgeability of LLMs about questions on their zero-shot question answering accuracy. We sample 10k questions with 1 unknown. We plot the distribution of knowledgeability $\bar{K}_{\min}$ (Equation 2) of Llama-3-8B-Instruct and Llama-3-70B-Instruct about

| Model | $\|\mathcal{G}\| = 2$ | $\|\mathcal{G}\| = 3$ | $\|\mathcal{G}\| = 4$ | $\|\mathcal{G}\| = 5$ | $\|\mathcal{G}\| = 6$ |
|---|---|---|---|---|---|
| Llama-3-70B-Instruct | 74.6 | 74.4 | 76.3 | 78.3 | 79.8 |
| Llama-3-8B-Instruct | 66.8 | 67.8 | 70.9 | 73.7 | 76.5 |
| Mixtral-8x7B-Instruct-v0.1 | 73.1 | 71.7 | 72.0 | 74.0 | 74.3 |
| gpt-3.5-turbo-0125 | 64.1 | 64.4 | 65.3 | 67.9 | 70.7 |

Table 9: Question answering accuracy with advanced retrieval, grouped by graph pattern size $\|\mathcal{G}\|$.

| Model | $\|\mathcal{G}_{\text{demo}}\|, n_{\text{unk\_demo}}$ | 4, 2 | 5, 2 | 6, 2 | 4, 2 | 4, 3 | 5, 2 | 5, 3 |
|---|---|---|---|---|---|---|---|---|
| | $\|\mathcal{G}\|, n_{\text{unk}}$ | 4, 3 | 5, 3 | 6, 3 | 5, 2 | 5, 3 | 6, 2 | 6, 3 |
| Meta-Llama-3-70B | | 29.8 / 29.8 | 27.5 / 30.7 | 31.5 / 30.4 | 50.6 / 51.9 | 31.4 / 31.6 | 53.4 / 56.3 | 29.1 / 30.5 |
| Llama-3-8B | | 16.3 / 18.0 | 14.4 / 17.1 | 20.9 / 21.4 | 33.3 / 37.8 | 21.7 / 18.9 | 30.1 / 39.0 | 8.0 / 9.3 |
| Mixtral-8x7B-v0.1 | | 23.8 / 23.9 | 22.8 / 21.4 | 23.4 / 20.8 | 47.4 / 47.3 | 26.1 / 24.8 | 48.7 / 50.0 | 17.4 / 20.8 |

Table 10: 4-shot and 8-shot CoT accuracy for questions generated from Wikidata with mismatched reasoning patterns between demonstrations and test questions (the graph pattern size, $\|\mathcal{G}_{\text{demo}}\|$ vs. $\|\mathcal{G}\|$, and the number of unknowns, $n_{\text{unk\_demo}}$ vs. $n_{\text{unk}}$). All open-source models are instruction-finetuned version.

| Model | $\|\mathcal{G}_{\text{demo}}\|, n_{\text{unk\_demo}}$ | 4, 2 | 5, 2 | 6, 2 | 4, 2 | 4, 3 | 5, 2 | 5, 3 |
|---|---|---|---|---|---|---|---|---|
| | $\|\mathcal{G}\|, n_{\text{unk}}$ | 4, 3 | 5, 3 | 6, 3 | 5, 2 | 5, 3 | 6, 2 | 6, 3 |
| Llama-3-70B | | 32.8 / 33.7 | 37.0 / 37.9 | 38.4 / 37.5 | 58.7 / 60.2 | 33.8 / 34.5 | 51.7 / 66.5 | 38.4 / 37.1 |
| Llama-3-8B | | 24.1 / 24.1 | 19.4 / 24.1 | 23.6 / 21.5 | 56.4 / 57.1 | 11.2 / 11.9 | 37.4 / 56.0 | 13.1 / 16.4 |
| Mixtral-8x7B-v0.1 | | 29.3 / 27.0 | 31.7 / 31.2 | 31.1 / 30.3 | 63.7 / 61.6 | 27.2 / 26.6 | 53.8 / 65.6 | 29.8 / 30.1 |

Table 11: 4-shot and 8-shot CoT accuracy for questions generated from T-REx with mismatched reasoning patterns between demonstrations and test questions (the graph pattern size, $\|\mathcal{G}_{\text{demo}}\|$ vs. $\|\mathcal{G}\|$, and the number of unknowns, $n_{\text{unk\_demo}}$ vs. $n_{\text{unk}}$). All open-source models are instruction-finetuned version.

questions that they answer correctly and incorrectly in Figure 5. We find that although both LLMs tend to be more knowledgeable about questions that they answer correctly than those they answer incorrectly, a considerable overlap exists between the distributions. Overall, these results suggest that

**Finding 4**: LLMs' confidence (as measured by log-likelihood) in their answers reflects poorly their confidence in the underlying knowledge (as measured by log-likelihood).

In practice, LLMs' completions may not start with entity names or "Yes"/"No" immediately, making it hard to extract the necessary log-likelihoods. To solve the problem, we use system prompts and few-shot demonstrations to make LLMs utter the tokens immediatetly.

## 5 RELATED WORK

**LLMs for Data Generation** LLMs have been widely used for data generation and augmentation. Self-Instruct Wang et al. (2022) and Evol-Instruct Xu et al. (2024) utilizes a small set of human-written tasks and instructions to seed LLMs for generating a large number of samples for new tasks. Orca Mitra et al. (2023); Mukherjee et al. (2023) incorporates rich reasoning signals, such as explanation traces and step-by-step thought processes, to enhance the model's reasoning capability. GLAN Li et al. (2024) and LAB Sudalairaj et al. (2024) leverage taxonoy-guided data generation by decomposing human knowledge and capabilities into hierarchical structures.

**LLM reasoning** LLMs have demonstrated strong reasoning abilities. Many works Wei et al. (2022); Yao et al. (2023); Zhou et al. (2023); Luong et al. (2024); Hao et al. (2023) have proposed various methods to enhance the reasoning capabilities of LLMs, incorporating finetuning, in-context learning and advanced prompt engineering techniques. In addition to these methods, there are numerous benchmarks Cobbe et al. (2021); Hendrycks et al. (2021); Zellers et al. (2019); Sawada et al. (2023) specifically designed to evaluate different aspects of LLMs, such as math reasoning and commonsense reasoning.

> What is the headquarters location of the employer of someone who was born in Sacramento and educated at Occidental College?
> Answer: Joe Rohde, born in Sacramento, educated at Occidental College, works at Walt Disney Imagineering, which is headquartered in Glendale. So the answer is Glendale.

Figure 4: Example chain-of-thougts demonstration (green) generated by Llama-3-70B-Instruct.

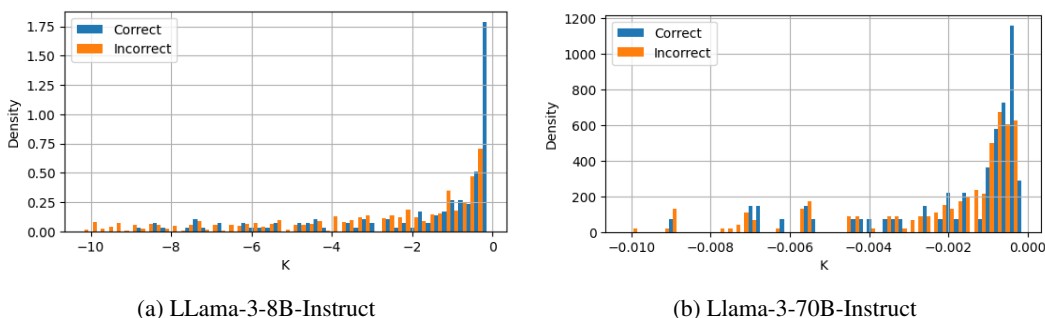

(a) LLama-3-8B-Instruct           (b) Llama-3-70B-Instruct

Figure 5: Distribution of knowledgeability for incorrectly answered questions, ploted side by side: although both LLMs tend to be more knowledgeable about questions that they answer correctly than those they answer incorrectly, a considerable overlap exists between the two distributions.

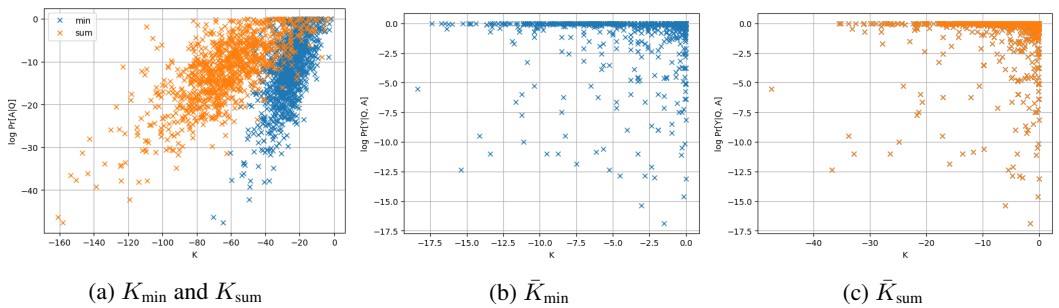

(a) $K_{\min}$ and $K_{\text{sum}}$       (b) $\bar{K}_{\min}$       (c) $\bar{K}_{\text{sum}}$

Figure 6: Knowledgeability transparency: the correlation between the LLM's confidence in its answers, and its knowledgability about the questions with different measures.

**Knowledge probes** Probing techniques have enabled numerous studies on the storage, retrieval, and editing of knowledge within LLMs. They play an important role in studying the internal representations and behavior of LLMs, helping in building more interpretable LLMs. These studies, as documented in references Zhang et al. (2024); Allen-Zhu and Li (2024); Meng et al. (2023); Gurnee and Tegmark (2024), enhancing our understanding of LLMs.

**Uncertainty quantification.** Uncertainty quantification Abdar et al. (2021); He and Jiang (2024) has been an active research area for developing more robust, reliable and trustworthy LLMs. Typical uncertainty quantification methods include confidence-based methods Hu et al. (2023) and conformal prediction Ye et al. (2024); Quach et al. (2023), showing significant promise in enhancing model reliability and interpretability.

# 6 CONCLUSION

To address the limited understanding of LLMs' reasoning capability in domains requiring substantial real-world knowledge, we introduce a novel pipeline designed to automatically generate realistic, knowledge-intensive question-answering benchmarks that necessitate intricate reasoning skills. By leveraging open knowledge graphs, the graph query language SPARQL, and LLMs, our pipeline eliminates the need for manual annotation, enabling scalability to unprecedented benchmark sizes and knowledge coverage. The benchmarks generated by our pipeline are then used to evaluate several state-of-the-art LLMs, revealing their susceptibility to errors and even hallucinations. Despite attempts to mitigate these issues through techniques such as retrieval-augmented generation and chain-of-thoughts prompting, their effectiveness remains limited. Our unique benchmarks also facilitate an examination of how well LLMs' confidence in their reasoning outcomes aligns with their confidence in the underlying knowledge - a pioneering study in this field, to the best of our knowledge.

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
