# OpenReview forum: "Assessing the Knowledge-intensive Reasoning Capability of Large Language Models with Realistic Benchmarks Generated Programmatically at Scale"
_ICLR.cc/2025/Conference — Submitted to ICLR 2025_

### Official Review · Reviewer_3p7V · 2024-11-03

**Soundness:** 3
**Presentation:** 3
**Contribution:** 3
**Rating:** 6
**Confidence:** 3

**Summary:**

This paper proposes a novel pipeline that automatically generates knowledge-intensive QA benchmarks with knowledge graph. The pipeline avoid manual annotations, so it can scale to a large size with diverse knowledge coverage. With this benchmark, the paper evaluates several SOTA LLMs and finds that LLMs is still hard to leverage world knowledge for reasoning. Also, the paper also find that LLMs is poorly confident to themselves in the underlying knowledge, which is valuable for the future research.

**Strengths:**

1. This paper utilizes KG, SPARQL and LLMs to programmatically generate QA benchmark, which is reasonable and can be transferred to another domain or scenarios.

2. This paper use the open source KG T-REx and Wikidata to build the benchmark, ensuring that the factual knowledge is factual knowledge involved in the question is included in the pre-training phase of LLM. Therefore, the experimental results and analysis on LLM reasoning is valuable.

3. The experiments include many basic methods for enhancing LLMs reasoning, which is sufficient.

**Weaknesses:**

1. Some of the claims in the paper are lacking reference. For example, "We choose SPARQL because we find state-of-the-art LLMs,such as GPT-4, demonstrates strong capabilities of translating SPARQL queries into natural language questions."(Lin140-142). It would be better to add some reference about this claim.

2. Some experiments result are lacking baseline, such as GPT-4o in Table 8-11. It would be better to claim why you don't conduct experiment on it.

**Questions:**

The knowledge transparency in Section 3 is confusing. Here are some of my questions:

1. In the **limitation and alternative**, why you conclude that log Pr[a|q,M] = 1 > K_M(q,a)? Pr[a|q,M] is less or equal than 1 so I think the equation is wrong.

2. Why you propose K_sum(q,a) and K_min(q,a)? How does it measure the quality of the reasoning ability of an LLM?

3. What's the motivation of replacing log Pr[a|q,M] with log Pr[Y|q_a,M]?

---

> ### Author Response · Authors · 2024-11-22
>
> Thank you for your feedback. To answer your questions:
> 1. This is a typo and should be Pr[a|q,M] = log(1) > K_M(q, a) instead.
> 2. K_sum and K_min represent respectively the average and minimum knowledgeability of a model about all facts required to correctly answer a question. Both only measure the amount of knowledge possessed by LLMs rather than their reasoning ability.
> 3. The motivation of replacing log Pr[a|q,M] with log Pr[Y|q_a,M] is that Pr[a|q,M] can make models appear more knowledgeable than it actually is. An illustration of this can be found in Line 207-214.

---

### Official Review · Reviewer_zzHe · 2024-11-03

**Soundness:** 2
**Presentation:** 1
**Contribution:** 2
**Rating:** 5
**Confidence:** 5

**Summary:**

This paper discusses an automated approach to creating benchmark datasets from knowledge graphs, as an alternative to traditional LLM QA benchmarks that require manual labeling. The proposed novel pipeline includes generating SPARQL queries from subgraph patterns within the knowledge graph, followed by converting these SPARQL queries into natural language questions. This approach emphasizes the ability to produce reliable benchmarks without human intervention, as it allows for direct comparison between the deterministic answers retrieved via SPARQL queries and the responses generated by the LLM. Experiments were conducted to observe how current state-of-the-art LLMs perform on the proposed benchmark.

**Strengths:**

The overall idea of the paper is easy to understand and intuitive. The authors' argument for the necessity of automatically generating QA benchmark datasets to evaluate LLMs’ reasoning capability is reasonable. Additionally, the authors conducted various experiments, and the process of deriving meaningful findings from these experiments appears appropriate.

**Weaknesses:**

1. The English in this paper could be refined for clarity. There are several minor and major typos throughout, and the figures and tables need to be arranged more appropriately to improve readability.
2. While it is understood that this paper’s programmatic generation pipeline has merit, as traditional QA benchmarks involve human effort, there is a lack of explanation and experimentation on how the quality of these automatically generated benchmarks compares to others.
3. The accuracy evaluation of the translated natural language queries presented in Table 3 appears valuable. However, it would be helpful to include more concrete examples of the natural language queries generated from SPARQL queries containing 1, 2, or 3 unknowns for better understanding. There also seems to be a lack of verification on what criteria were used to judge the correctness of the natural language translations for each SPARQL query, especially in terms of assessing the "naturalness" of the language used. The question shown in Figure 3 appears somewhat awkward, suggesting that the naturalness of queries may decrease as the number of unknowns increases.
4. Although the paper presents many impressive experiments and multiple result tables, there is a lack of in-depth analysis and interpretation of each result. To derive the four findings proposed in the paper, more detailed analysis about experimental results is necessary.
5. The proposed four findings do not seem particularly novel, as they closely resemble well-known findings in the field.
6. The purpose of this study is focused on the automatic generation of benchmarks from KGs, and it is understandable that the paper includes experiments to evaluate these benchmarks. However, the logical progression and structure of the paper feels unclear. Testing with RAG and CoT appears more like testing the performance of SOTA LLMs than demonstrating the quality of the proposed benchmark pipeline.
7. In the "Methodology Limitations" section, it is stated that the performance of LLMs is similar between benchmarks created from T-Rex and Wikidata, despite large differences in triple counts, implying that ground truth incompleteness is not a significant issue. However, this claim is not entirely convincing, as the difference in triple counts may not be directly related to the issue of ground truth incompleteness.

**Questions:**

Compared to existing benchmarks, in what ways does the proposed benchmark provide advantages beyond the ability to be generated automatically? Could the authors elaborate on specific aspects where this benchmark may offer superior quality or unique insights?

---

> ### Author Response · Authors · 2024-11-22
>
> Thank you for your feedback. To answer your question about the advantages of our benchmark, a key advantage of our benchmark is its improved knowledge coverage. More details about this can be found in Section 2.3.

---

> ### Comment · Reviewer_zzHe · 2024-12-01
>
> Thank you for your feedback. Section 2.3 highlights the thoughtful selection of knowledge graphs that align well with SOTA LLMs for benchmark generation. While T-REx and Wikidata are indeed reliable and widely adopted resources, their use might not substantially differentiate the proposed benchmark in terms of its novelty. Perhaps elaborating on how their specific integration enhances the benchmark’s unique qualities could strengthen the argument.

---

### Official Review · Reviewer_gnpn · 2024-11-03

**Soundness:** 2
**Presentation:** 2
**Contribution:** 2
**Rating:** 5
**Confidence:** 3

**Summary:**

The paper presents a new benchmarking technique for LLMs for knowledge intensive applications. The authors use KGs and SPARQL to generate subgraphs and then use LLM to generate NL questions, the authors use these questions to benchmark different LLMs in different settings(zero shot, RAG and CoT). The pipeline to generate benchmarks is automated and thereby this is a scalable approach. The authors generate benchmark datasets and benchmark many LLMs.

**Strengths:**

1. Using SPARQL and Knowledge graphs to generate benchmark questions is a novel idea. The benchmark pipeline can definitely be used to generate complex benchmark datasets.
2. Since this is completely automated, the approach is scalable and thereby one can generate significant number of benchmark datasets which is specific to the use case and LLM needed to be benchmarked.
3. The authors also have done a good job of explaining the limitations of the work and have clearly mentioned their focus area for this paper.
4. The experimentation is well done - The authors tested the LLMs on three major LLM settings(zero-shot, RAG and CoT). The complexity(number of unknowns) of questions was taken into consideration for experimentation.

**Weaknesses:**

1. Data Quality of the generated NL questions. The only quality check the authors have done is manual review. The number of questions seems low(1200). However, the methodology used to select the manual review questions is only based on number of unknowns. Potentially, the authors could have looked at question types and selected questions accordingly. It is possible that LLMs are predominantly bad at converting certain kind of sparql queries. So the LCB might need further evidence.
2. The quality of the approach is in general limited by LLM used to convert queries to NL.
3. The authors could have presented more evidence about how this benchmark dataset is better than existing benchmarks mentioned in the paper. Are the quality of questions better than template and manual questions curated?

**Questions:**

1. In section 2.2, there is a claim "We choose SPARQL because we find state-of-the-art LLMs, such
as GPT-4, demonstrates strong capabilities of translating SPARQL queries into natural language
questions." - Is there a reference missing here? To what extent is GPT4 good at converting sparql queries to NL?
2. Is there evidence of how the benchmark itself if better than existing template based and manually curated benchmarks available?
3. Does higher volume of questions help in any meaningful way to benchmark datasets? If yes, is there evidence of that?

---

> ### Author Response · Authors · 2024-11-22
>
> Thank you for your feedback. To answer your first question, the claim is based on our manual evaluation of natural language questions converted by GPT-4 from SPARQL queries. To answer your second and third question, the volume of questions lead to improved knowledge coverage compared to manually curated benchmarks since we deduplicated our questions.

---

> > ### Comment · Reviewer_gnpn · 2024-12-03
> >
> > Thank you for the response. I understand that the volume of questions after deduplication might lead to improved knowledge coverage. I was curious if there is any study on diversity of questions.

---

### Official Review · Reviewer_zdNL · 2024-11-05

**Soundness:** 2
**Presentation:** 2
**Contribution:** 2
**Rating:** 5
**Confidence:** 3

**Summary:**

Current QA benchmarks, like, HotpotQA rely on human annotations. This limits their scale and coverage making it prohibitively expensive to expand the benchmarks. This motivates the need for an automated pipeline for benchmark curation that is scalable both in terms of size and coverage.
To that end the authors propose a pipeline that leverages existing knowledge graphs (KG). They sample subgraphs from the KG, mask certain number of entities and create SPARQL queries for the masked subgraphs. These are then translated into natural language questions by an LLM and the corresponding ground truth is obtained by querying the KG. They generate a benchmark of size 1.32M which is an order of magnitude larger than existing benchmarks. They then evaluate existing SOTA LLMs on their benchmark and draw insights on the current knowledge-based reasoning capabilities of LLMs.

**Strengths:**

With the progressive evolution of LLMs, designing robust evaluation benchmarks is the need of the hour. Reasoning is one of the important capabilities of LLMs and there is a large body of work covering the taxonomy of reasoning, approaches, and evaluation. The authors address an important problem that stands to have a wide impact on the LLM research community.
The idea of leveraging knowledge graphs and SPARQL for intermediate representation is neat and the authors develop this into an elegant pipeline for automated curation of knowledge-based reasoning benchmark.
The generated benchmark dataset is fairly large - almost an order of magnitude larger than the existing benchmarks - thereby justifying the initial motivation.
Through evaluation of the current top LLMs on their benchmark, the authors highlight key insights / gaps on their performance on knowledge reasoning. These insights could help drive forward the research on improving reasoning in LLMs.

**Weaknesses:**

One of the motivations cited is the lack of sufficient coverage in the existing benchmarks. Yet the authors do not provide details on how their benchmark addresses this. I would have expected details on topic coverage or similar. How are these knowledge graphs created? Are they created manually or through curation pipelines from unstructured knowledge bases? If so, how well do they cover the underlying knowledge? A lot of these questions are left unanswered.
The pipeline involves several steps. The authors should provide evaluation on individual steps, ablations, highlight challenges. For instance, the authors highlight challenges with customised domain-specific languages. However, there are no relevant citations nor evaluation on why SPARQL is a better choice as an intermediate representation.
While the authors report accuracy of current LLMs, it is unclear if the gaps are with information recall, reasoning, hallucination or other. The authors present some anecdotal examples but some quantitative details would make the insights stronger and actionable.

**Questions:**

"Less labor intensive approaches often rely on customized domain-specific language as intermediate representation (IR). Constructing such IRs itself can also require significant manual efforts. Additionally, customized IRs may not generalize well to novel knowledge."
Missing citation

---

> ### Author Response · Authors · 2024-11-22
>
> Thank you for your feedback. Section 2.3 contains information about how the KGs that we use (Wikidata and T-REx) were created and the amount of knowledge that they cover (measured by number of triples). Several papers cited by our paper, for example, ComplexWebQuestions (Talmor and Berant, 2018), offer insights into challenges faced by approaches that rely on customized domain-specific languages.

---

> > ### Comment · Reviewer_zdNL · 2024-11-26
> >
> > Thank you for the response. The paper should include these details.

---

### Meta-Review · Area_Chair_waTB · 2024-12-21

**Metareview:**

This paper proposes a pipeline that automatically generates knowledge-intensive QA benchmarks with knowledge graphs. The reviewers are positive about the experiments and presentation. However, there are also concerns including the coverage of existing benchmarks, quality/diversity of the questions, and novelty of the proposed benchmark. Some of these concerns were not fully addressed during the rebuttal. Overall, I think the paper can be improved and should go through another round of reviewing.

**Additional Comments On Reviewer Discussion:**

Main remaining concerns:
- Coverage of existing benchmarks: the revised paper should include more details.
- Quality/diversity of the questions: there should be a study on the diversity of questions as well.
- Novelty of the proposed benchmark: Using T-Rex and Wikidata may not be sufficient for the novelty.

---

### Decision · Program_Chairs · 2025-01-22

Reject